# Economic Contributions of Mega-Dam Infrastructure as Perceived by Local and Displaced Communities: A Case Study of Merowe Dam, Sudan

**Al-Noor Abdullah [1], Sanzidur Rahman [1,2,]*, Stephen Essex [3] and James Benhin [1]**

[1] Plymouth Business School, University of Plymouth, Plymouth PL4 8AA, UK;
al-noor.abdullah@plymouth.ac.uk (A.-N.A.); james.benhin@plymouth.ac.uk (J.B.)

[2] Faculty of Economics, Shandong University of Finance and Economics, Jinan 250001, China

[3] School of Geography, Earth and Environmental Sciences, University of Plymouth, Plymouth PL4 8AA, UK;
stephen.essex@plymouth.ac.uk

*   Correspondence: srahman@plymouth.ac.uk

**Abstract:** Investigations on the socioeconomic impacts of mega-dam projects have tended to focus on conventional cost–benefit analysis, while studies exploring perceptions of local communities, who are some of the prime beneficiaries of these development initiatives, are limited. This paper aims to address this research gap through a case study of community perceptions on the socioeconomic impacts of the Merowe Dam in Sudan from the residents of upstream, downstream, and relocated locations. Data were collected primarily through surveys and interviews with residents, government officials, dam implementation authority, non-governmental organizations (NGOs) and other key informants and a series of indicators were developed for the analysis from the responses. Three inter-related areas of impact were scrutinized: (a) electricity generation; (b) development of modern agriculture; and (c) institutional infrastructure in the region. The results reveal that local communities are fully aware of both the positive and adverse socioeconomic impacts of the Merowe Dam, although these are focused more on the visible impacts closely related to their livelihood and income, such as increased food production, water shortages, electricity supply and its costs. Policy implications include investments in the new settlement areas with respect to the agricultural economy, such as irrigation improvement through electrification, promoting crop diversity, research, development, and diffusion of modern agricultural technologies. Efforts are also needed to strike a balance between provision of utilities and services, (i.e., water, electricity and other infrastructural facilities) provided by the Merowe Dam, amongst communities in relocated, upstream, and downstream locations.

**Keywords:** community perception; economic impact; electricity generation; modernization of agriculture; socioeconomic development and institutional infrastructure

## 1. Introduction

Global economic development has been undergoing a transition and in developing countries, a shift from traditional development to modern economic development has taken place since the 1980s [1,2]. Many developing countries have used mega-dams as a source to generate electricity, water for irrigation, and to facilitate urban and industrial uses. Mega-dams research has covered many areas such as aversion of climate change, corruption related to political involvement, ideology/symbols of state building, social effects examining displacement and relocation and economic effects in relation to livelihood, electricity, and infrastructure for development [3–6]. The economic investigation mainly emphasized cost–benefit analysis, while studies exploring perceptions of local communities, who are perhaps some of the prime beneficiaries, are limited. Perception of the economic impacts of mega-dams

by affected communities is important since people's views encompass achievement of goals, including those already achieved and those yet to be achieved and, therefore, might be considered as a guiding concept of behavior and/or decision-making. The socioeconomic impacts of mega-dams center around the displacement and relocation of communities, damage to livelihoods, loss of income sources, and health hazards to affected communities [7–9]. As Welzel et al. [10] suggested, any economic change in a society, such as dam construction, may have a profound impact on peoples' livelihoods, positively or negatively, which reflects their socioeconomic conditions.

These factors are closely monitored in investigating socioeconomic and human development. Socioeconomic positions of individuals and groups depend upon a combination of factors, such as occupation, education, income, wealth, place of birth, residence, etc. These aforementioned factors are influenced by the development of a mega-dam, and whether the impact of each factor is positive or negative, depends on the type of communities and factors investigated (livelihood, loss of income, and economic and social impacts on resettled and displaced communities) [6,11,12]. The present study aims to address the effects of mega-dam development at the community level and the underlining issues related to the daily lives of the ordinary people as perceived by the people themselves. The significance of 'micro-level' assessment applied in this research allows the researcher to avoid the shortcomings of relying heavily on macro-level and large numeric data, which either ignores or masks impacts of such mega-dam projects at the grassroots and/or community levels where such mega-infrastructure is physically located. Hence, the study focuses on providing a detailed analysis on how mega-dams influence delivery of socioeconomic development at the community level based on the diagram presented in Figure 1.

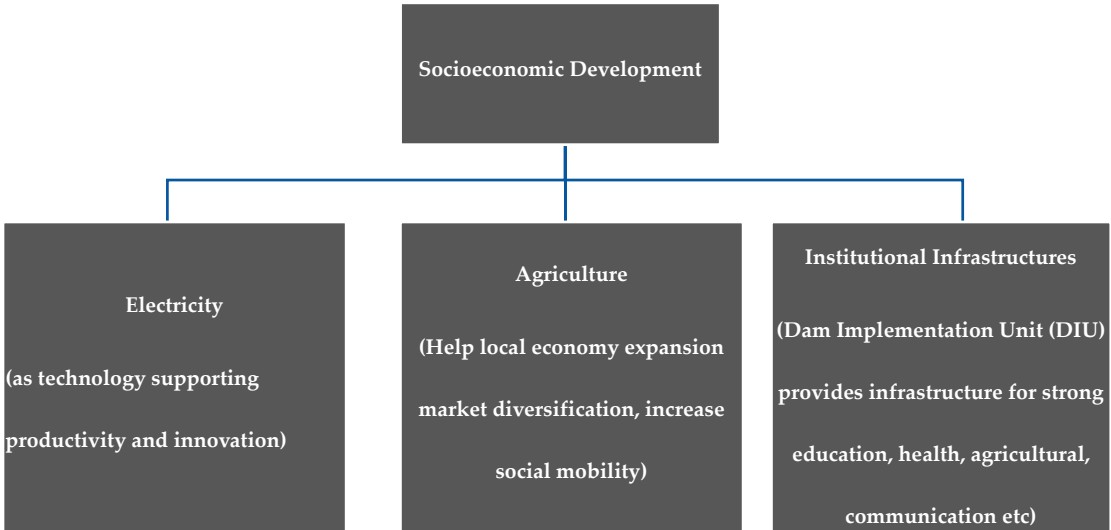

**Figure 1.** Conceptual framework for analyzing socioeconomic theory. Source: Author.

Although previous researches thoroughly investigated the economic and social impacts of dams, the issues of funding stagnation and subsequent return of mega-dams into the global agenda with the emergence of China and Brazil [13], assessment of the perceptions of locals has received limited attention, despite integrity of socioeconomic aspects being an essential prerequisite for the development of sustainable communities [14]. This reflects rapid changes in institutions and initiatives in and around rural, local, national, and international contexts that may influence developing economies in their quest for electricity needed for social and economic modernization [15–19].

Studies that analyzed mega-dams to some degree have focused on the macro-economic factors of dams' impact, applying cost–benefit analysis and quantitative methods relying on standard large data with less emphasis on the context of local communities and perceptions of those affected [20,21]. Furthermore, less emphasis was placed on the agency of society, economic structure, development stage,

needs of the economy, geopolitics, level of institutional development, and sociocultural factors [22,23]. However, some studies conducted by scholars such as Baviskar [24], Everard [25], Tischler [26], Isaacman and Isaacman [27], and Biswas et al. [28] on communities' perceptions suggest that local people's perceptions of large dam infrastructure, for good reason, tends to be negative. This perception is due to negative impact dams might have on many socioeconomic factors of local communities ranging from displacement to loss of livelihoods. However, this does not reflect the overall outcome of any given dam project as the degree of negative impact might differ in addition to the positive impacts that dams might have on local communities. Therefore, the argument that dams create negative impacts seems to be extreme and disregards many benefits gained by communities in many countries, such as, Nepal, Turkey, and other areas [23,29].

The debate over large-dams' socioeconomic impact is becoming ever more polarized especially in recent years. The polarization appears to reflect the frustration of those with opposing viewpoints, rather than spurring a useful dialogue between stakeholders around the challenging choices related to managing water, energy, and other natural resources [19,30]. Therefore, homogenizing the socioeconomic impact of dams can undermine many socioeconomic development programs needed to support the improvement of communities in different parts of the world [23].

Electricity generation, related institutional infrastructure, and irrigation to support agriculture, are the main aspirations of governments and communities behind dam construction. Critics perceive this aspiration to be politically motivated due to unequal participation of affected communities in decision-making and injustices dispensed on the displaced, especially in countries governed by authoritarian regimes [4,16,20,23,31–34].

The present study focuses on the analysis of economic contributions of dam projects encompassing agricultural and non-agricultural (i.e., electricity and institutional infrastructure) development, which are usually the main justifications of undertaking mega-dam projects worldwide [15,16,18,23]. This economic justification is an area of debate in the literature, where different views, thoughts, claims, and counterclaims on the economic contributions of dams exist [31,35,36]. This study pursues the lines of a contribution to the literature on benefit sharing from dams, its potential, what works and what does not, and for whom do benefit sharing schemes exists, such as electricity, infrastructure, agricultural, and irrigation schemes [15,16,18,23]. This is an area receiving great academic attention on its effectiveness, and what is being learned (or not) about dams' multifaceted impacts.

To understand the debate on mega-dams' socioeconomic impact, it is important to measure their outcomes against a framework. Therefore, many studies have used different theories, such as political ecology, modernization, and/or sustainable development to examine the effect of dams on the hosting community, region, and country levels [37]. In this study, socioeconomic development theory is used to investigate economic changes brought about by Merowe Dam. The motive behind selecting this theory is to provide a framework that allows the researcher to examine the economic impacts at the affected community level, which is interrelated with social and political factors of the dam in the Merowe region. As Jaffee [6] suggests, understanding economic or social phenomenon is an extremely complex process. This complexity is due to the variation in factors (i.e., social, economic, political and environmental) involved in a dam project [6]. Therefore, having a clear understanding of how socioeconomic theory shapes peoples' interpretations and perceptions of an event is important. In effect, socioeconomic theory is important to address economic and social changes that occur in a region due to the construction of mega-dam, and its direct implications on human development in a region.

This research applied the three paths of socioeconomic theory from Welzel et al. [10] to address the Merowe Dam's impact on communities. These three paths are relevant in the context of the Merowe Dam as technology has had a huge impact on the lives of communities in the Merowe region, both socially and economically. This view is supported by many scholars including Marsh [37], Rigg [38], and Sen [39].

The first trajectory is the factors associated with technological innovation changes that dams bring (i.e., productivity, income, services, livelihood), which are linked directly and indirectly to socioeconomic development. The second trajectory is indirectly connected to dam development through expansion of markets and social mobilization due to increased income and availability of disposable income (i.e., more choices, luxury goods, improved houses, changes in lifestyle), which bring changes to the socioeconomic status of individuals and communities. Subsequently, human development prevails as economic activities and social mobility increase [10]. The third trajectory is based on societies interacting with the state institutions tasked with socioeconomic development (i.e., dam implementation authority and the government) in the form of consultation and compensation processes. The third path is also connected with institutional infrastructure related to complementary projects, including schools, colleges, health facilities, and other institutions supporting agricultural and non-agricultural activities in the region. In respect to this, greater changes were detected in states with strong established democracy [10]. In effect, the first two paths are linked to technological interaction with the market, to some degree, which led to socioeconomic improvement. The third trajectory corresponds with Amartya Sen's "capabilities theory" that enables communities to pursue fulfilling lives and have choices in development approaches pursued where governments enable societies and people to have "capabilities" to function effectively [37,38].

Given this backdrop, the study focuses on Sudan, which is one of the most vulnerable countries in terms of socioeconomic development. The Dam Implementation Unit (DIU) and the government of Sudan went to a great length to construct the Merowe Dam primarily aimed at contributing significantly to electricity generation for economic modernization. This modernization is closely linked with the development of institutional infrastructure to alleviate social and economic stagnation in the Merowe region through various supportive projects. The community perceptions are substantiated and validated by examining changes brought into three main areas of economic development identified by the communities: (a) electricity generation, (b) institutions and other infrastructure, and (c) agriculture using a wide range of relevant indicators (Figure 1).

## 2. Methods

### 2.1. Study Area: The Merowe Region

The case study area is the Merowe region situated in the northern state of Sudan. The landscape of the region consists of hills, flat desert, and mountains. Situated in a desert dry climate, this region has an average temperature of 46 °C, and in winter, the medium temperature falls to about 20 °C in January [40,41]. The area is only habitable on the fertile strip of land by the Nile and historically is known to be a very hostile place for living.

It is important to recognize the huge impact that the Merowe Dam has had on upstream and downstream communities, while not ignoring the economic and social benefits it has brought to these communities and the wider nation [15,16,23,42]. Previously, the entire Merowe region was isolated. Communities lived in small farming villages near the banks of the Nile and on islands [43]. Between 30,000 to 50,000 people were affected by the construction of the Merowe Dam and its reservoir from 2006 to 2009. This fallout from the dam mainly affected Manasir, Hamdab, and Amri communities (through displacement, and loss of culture, livelihoods, and date palm trees). These communities were selected to assess the impact the dam has had on them economically, socially, politically, and environmentally [44]. It is important to assess the impact of the Merowe Dam on farming at both new and old settlements. Commonly, researchers study the impact of dams on upstream communities and the displacement of people by the reservoir [45,46]. However, this research also investigates downstream communities to study the changes that the new water regime has brought to the Hamdab West, Al Degawit, and Nouri communities and how this has influenced farming and the size of land cultivated.

Downstream communities are often affected due to changes in water flow variation, which affects farm production, fishing communities, and ecosystems [45,46]. This focus suggests not only

environmental impacts, but also economic and social impacts as well [47]. The selection of the above communities helped this research to identify and assess socioeconomic change brought on by the Merowe Dam. As Welzel et al. [10] suggested, any change in the status quo of these upstream and downstream communities, socially or economically, may have profound implications for people's livelihoods, which then reflect on their socioeconomic and human development. The Merowe region is regarded as a social, economic, administrative, and eco-spatial vessel that appropriately captures changes that researchers need to explain and clarify [38].

Case study is an important approach in social research. Yin [48] defines case study research as "An empirical inquiry that investigates a contemporary phenomenon within its real-life context, especially when the boundaries between phenomenon and context are not clearly evident" (p. 35). Some scholars have suggested that case study can be applied to any type of research, whether quantitative, qualitative or multi-methods [48–50]. Yin [48] further comments that the case study "comprises an all-encompassing method—covering the logic of the design, data collection techniques, and specific approaches to data analyses" (p. 35). The Merowe Dam was selected as a case study for being a meeting point for different forces and processes (economic, social, state power, environmental, and cultural changes) [51]. The fundamental idea behind the Merowe Dam as a case study was that the area is a unit within the wider context of North Sudan possessing unique features of economic, social, and cultural borders (see Figure 2). Other reasons for selection include representativeness of historical events of Aswan High Dam and it impact in the region, and the anticipated impact of proposed dam projects, such as Kajbar. Additional reasons why the Merowe Dam was chosen as the study site can be summarized as follows:

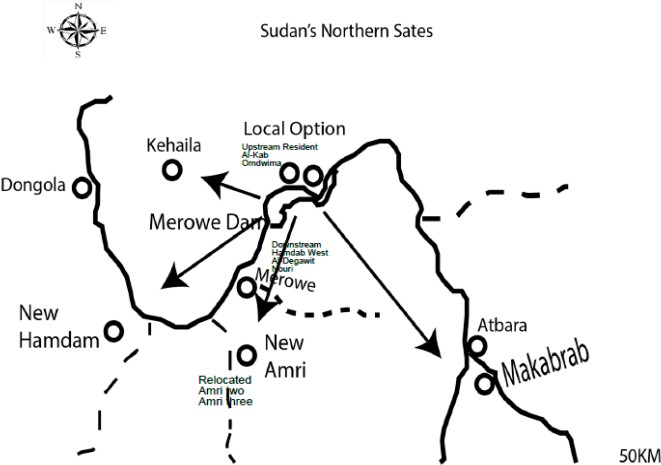

**Figure 2.** The Location of the study upstream and downstream. Source: Author.

There have been dramatic changes in government policies towards international investment in mega-projects. The historical context of influential mega-dams in this region, from the Aswan High Dam to Merowe Dam may recall sad memories of displacement for the communities. The dam has huge social and environmental impacts and stimulates major ideological, economic, and political debate in the regional and global contexts. The construction of the project started in 2004, went into operation in 2009, and became one of the largest mega-projects in the Nile-based countries [12]. The project was at the center of a political and economic war between countries from upstream to downstream locations, namely Ethiopia and Egypt, especially with the recent development of the Grand Ethiopian Renaissance Dam. The dam had a large impact on water flow downstream, which made this case study very compelling.

## 2.2. Data Collection

The lead author resided in the Merowe region for four months from April to September 2017. The research adopted a mixed-methods approach, which combined both quantitative and qualitative methods. First, the area was divided into three localities in the northern regions of Sudan: upstream relocated, upstream residents, and downstream residents (see Figure 2). The area impacted by the Merowe Dam consisted of over 100 villages and islands; the majority of villagers were relocated to four main settlements, but some displaced communities refused to relocate and preferred to stay by the reservoir. The selection of the seven villages was mainly guided by the need to ensure representation of displaced communities including those who were relocated and those who refused to relocate and stayed by the dam. The study focuses on the dam's impact on different locations. This led to selection of three villages located downstream where no displacement occurred or had greater impact in order to draw a comparison among the three types of locations. Samples from seven villages were collected; these included the upstream-relocated villages of Amri two and Amri three, downstream villages of Hamdab West, Al-Degawit and Nouri, and upstream-residents by the reservoir area Al-Kab and Omdwima. A total of 300 households out of 30,000 in the sample frame was selected for data collection following a multistage cluster random sampling procedure (100 per-locality divided equally between villages). Then, each of the seven villages was divided into four individual groups according to differences in landscape and socioeconomic characteristics. The 300 sample was also informed by the equation in Equation 1 as suggested by Becker and Murphy [52]. Equation 1 used to estimate the sample size. Based on the equation, the sample was estimated at about 264, but extended to 300 to compensate for potential non-response rates.

$$n = \frac{Nz^2 p(1-p)}{Nd^2 + z^2 p(1-p)} \tag{1}$$

where:

$n$ = sample size
$N$ = total number of households (1000 per community)
$z$ = confidence level (at 95% level, $z$ = 1.96)
$p$ = estimated population proportion (0.5, this maximizes the sample size)
$d$ = error limit of 6% (0.06)

An in-depth questionnaire survey was administered within the target communities, adopting a cluster sampling strategy to avoid any geographical or socioeconomic bias. A total of 33 households was selected randomly from two villages: 34 from one village from downstream district and 50 each from four villages of both upstream districts.

The questionnaire survey focused on family demographic characteristics, land, property owned, and attitudes/feeling towards the impacts of Merowe Dam in relation to agricultural production, information on livestock, and income. Furthermore, a set of 17 specific economic development (agricultural and non-agricultural) indicators related to socioeconomic impacts of the dam were developed (please see questionnaire in the Supplementary S1). These economic indicators listed in the questionnaire were identified through a literature review of previous studies on the impacts of dams as well as indicators specific to the Sudanese context [6,10,53,54]. In the next stage, respondents were then asked to indicate their opinions on the Merowe Dam. The socioeconomic impacts of the Merowe Dam were ranked on a five-point scale ranging from 1 = strongly negative and 5 = strongly positive. This approach is a common method of collecting data, which means that it is easy to understand. It is appropriate for qualitative data, easy to draw conclusions based on the results and graphs prepared from the responses. The perception of participants on the economic contribution of Merowe Dam in the study area was determined through a detailed socioeconomic analysis of a total of 17 indicators (nine related to non-agricultural economic activities and infrastructure and the remaining eight related to agriculture,

see Tables 1–8). In addition, 33 key informants were selected for in-depth interviews from a range of actors, comprising six village community leaders and experienced farmers; 15 academics, officials of non-governmental organizations (NGOs) and civil society; nine government officials (ministers, policymakers and Merowe Dam authorities); two businessmen; and two focus group discussions with local communities and university students. This approach that is sensitive, person-centered and offer interviewees freedom to think and express their opinions as well as contrasting, supporting or correcting data from questionnaire respondents [49]. However, semi-structured or in-depth interviews are not free of problems. With interviewer bias, participants might offer the answers that they think the interviewer wants [49].

## 3. Results

The main aims of the Merowe Dam and its supplementary projects were to support agricultural and non-agricultural economic development through electricity generation and the construction of related institutional infrastructure to resolve the problems of agricultural stagnation. The supplementary projects are in the form of educational and health facilities, newly electrified irrigation system, paved roads, and bridges for transportation, internet and telephone networks, markets, etc., which remain the greatest constraints in Sudan to serve as a dynamic economy especially the agricultural economy. Northern States are relying heavily on agriculture economically, thereby indicating the significance of agro-economic development to improve socioeconomic conditions in the region. Table 1 presents the demographic information of the sampled respondents. The overall mean age was 37.5 years and mean education was 13.6 years which indicated that participant had education beyond high school (scores: university = 16; high school = 12; secondary = 9). The male-dominated distribution shows that the perceptions received in this study are mostly by male members. Most respondents were reasonably educated and, therefore, assumed to be well informed and have a sound judgement capability about the dam's impact. The domination of male respondents is due to cultural norms in Northern Sudan where women have limited involvement in public affairs, especially related to political and economic matters. Table 1 further shows that 73% of the participants were engaged in agriculture in some form, especially farming of date palms which is a sign of economic and social pride for Northern Sudanese people.

**Table 1.** Summary statistic of the demographic characteristics in three districts.

| Items | Unit of Measurement | DownStream | Relocated | Upstream | Total |
|---|---|---|---|---|---|
| Age of head | Years | 40.79 (11.34) | 36.85 (11.83) | 34.89 (10.66) | 37.51 (11.52) |
| Education of head | Completed years of schooling | 13.15 (3.36) | 13.56 (3.31) | 14.05 (3.19) | 13.59 (3.30) |
| Male respondent | Percentage | 64.0 | 73.0 | 74.0 | 70.3 |
| Female respondent | Percentage | 36.0 | 27.0 | 26.0 | 29.7 |
| Date palm farming | Percentage | 77.0 | 67.0 | 77.0 | 73.0 |
| Vegetable farming | Percentage | 62.0 | 60.0 | 48.0 | 57.0 |
| Arable crop farming | Percentage | 79.0 | 69.0 | 70.0 | 72.0 |
| Livestock farming | Percentage | 40.0 | 26.0 | 24.0 | 30.0 |
| Number of samples | | 100 | 100 | 100 | 300 |

Note: Figures in parenthesis are standard deviation. Source: Authors' calculation.

The analysis of participants' perceptions related to the agricultural and non-agricultural benefits brought about by the dam suggests that the Merowe Dam generated more winners (downstream and relocated communities and the country) and fewer losers (upstream-relocated and resident communities living by the dam). These perceptions were derived from a series of questions posed to the respondents (please see attached questionnaire in the Supplementary S1). Despite the criticism of the degree of benefit of the dam's construction, there has been some support from the World Bank for mega-dams mainly due to their benefit to communities and the environment, especially in the era of climate change [29]. Environmental criticism of dams seems to be weighed against windmill and solar power as the less harmful mean of generating electricity compared to coal and thermal power. The study also found that the agricultural and non-agricultural economic contributions of Merowe Dam

embedded within the overarching socioeconomic transformation of the region. The results suggest that the perceptions of local communities concerning the agricultural and non-agricultural economic impacts of Merowe Dam significantly vary regarding electricity supply, agriculture, and infrastructure development in the region (Table 2).

The differences in perception on various questions regarding the contribution of the Merowe Dam varies significantly across the location of the respondents instead of gender and/or educational level of the respondent (Table 3). This is because a cross-analysis of the same perception questions by gender of respondents and educational level of respondents shows no significant differences in opinion. In other words, the perceptions regarding the contribution of Merowe Dam significantly vary by location of the respondents and populations residing within the same individual location have the same type of perception irrespective of gender and educational levels. It also provides confidence in our results that the choice of three distinct type of locations—downstream, upstream relocated, and upstream resident by the reservoir—is appropriate to examine the multifaceted contributions of this mega-infrastructure project, where residents within each type of location experience the same effect, whether positive or negative, and irrespective of gender or educational level.

These results presented in Table 2 suggest that economic contributions of mega-dams are a point of disagreement, nationally and internationally, due to their apparent overstated benefits and high costs of construction. The critics of mega-dams believe that benefits of such projects are inflated in order to attract investors and garner public support [7,19]. As some critics suggest, Merowe Dam also exerts huge social and environmental coststo some degree inequality in benefit distribution to some degree, lack of participation in decision-making, and displacement [30,34]. Nevertheless, many countries (e.g., Turkey and India) have benefited economically from the construction of dams [23,31]. In the case of Merowe Dam, it can be argued that its main purpose was to generate electricity, provide irrigation, and stimulate other socioeconomic development. The hydropower is connected to irrigation through three huge agriculture schemes to support relocated communities in addition to private and government commercial agribusiness projects. Furthermore, the government has a plan to create two canals on the east and west bank of the dam for irrigation purposes, although lack of funds has caused a delay in implementing the plan. The government has built infrastructure to support agriculture and socioeconomic development in the region, such as new roads, airport, education, health institutions, and the Merowe Agriculture Development Authority. Most of these infrastructures are owned by the government and the farmers only own the land, but there are new players entering agribusiness in the region, such as Al Rajhi Group a Saudi Arabian company and other private local companies. Outsiders' involvement supports the claim of inequality of distribution of dams' benefits where big companies take advantage of such development at the expense of local small famers [34]. However, as the results suggest, most of the participants in the region are in favor of Merowe Dam and its associated projects and view it as a means to support socioeconomic development in the region. The results suggest that there are other benefits detected based on the communities' perception, although the importance of these benefits can vary considerably between communities (Table 2). The above analysis suggests that dams could be regarded as both harmful and beneficial for development and with more preparation and good planning, especially with displacement and relocation, the harm can be mitigated.

**Table 2.** Participants' perception on Merowe Dam's agricultural and non-agricultural economic benefits by location.

| Location | Electricity | | Irrigation | | Other Purposes | | | | Final Opinion about Merowe Dam | | | | Total |
|---|---|---|---|---|---|---|---|---|---|---|---|---|---|
| | No | Yes | No | Yes | Nothing | Economic Development | Human Development | No Answer | Economic Contribution | Electricity Contribution | Very Good for Regional Development | Failure | |
| Downstream | 7 | 93 | 31 | 69 | 60 | 28 | 12 | 13 | 47 | 3 | 27 | 10 | 100 |
| Upstream Resident | 9 | 91 | 30 | 70 | 69 | 21 | 10 | 8 | 25 | 6 | 24 | 37 | 100 |
| Upstream Relocated | 12 | 88 | 45 | 55 | 73 | 19 | 8 | 2 | 25 | 7 | 20 | 46 | 100 |
| Total | 28 | 272 | 106 | 194 | 202 | 68 | 30 | 23 | 97 | 16 | 71 | 93 | 300 |
| | 1.497 * | | 6.156 * | | 4.09 * | | | | 43.205 *** | | | | |

Note: *** = significant at 1% level ($p < 0.01$); * = significant at 10% level ($p < 0.10$). Source: Authors' calculation.

**Table 3.** Participants' perception on Merowe Dam's agricultural and non-agricultural economic benefits by gender and educational level.

| | Items | Electricity | | Irrigation | | Other Purpose of Merowe Dam | | | | Final Opinion about Merowe Dam | | | | Total |
|---|---|---|---|---|---|---|---|---|---|---|---|---|---|---|
| | | Yes | No | Yes | No | Nothing | Economic Development | Human Development | No Answer | Economic Contribution | Electricity Contribution | Very Good for Regional Development | Failure | |
| Gender | Male | 188 | 23 | 139 | 72 | 140 | 49 | 21 | 12 | 71 | 10 | 49 | 69 | 211 |
| | Female | 84 | 5 | 55 | 34 | 61 | 19 | 9 | 11 | 26 | 6 | 22 | 24 | 89 |
| | Total | 272 | 28 | 194 | 106 | 201 | 68 | 30 | 23 | 97 | 16 | 71 | 93 | 300 |
| | Chi-squared | 2.064 ns | | 0.456 ns | | 0.142 ns | | | | 5.210 ns | | | | |
| Education Level | Secondary | 32 | 2 | 23 | 11 | 26 | 5 | 3 | 1 | 12 | 1 | 4 | 16 | 34 |
| | Higher | 240 | 26 | 171 | 95 | 175 | 63 | 27 | 22 | 85 | 15 | 67 | 77 | 266 |
| | Total | 272 | 28 | 194 | 106 | 201 | 68 | 30 | 23 | 97 | 16 | 71 | 93 | 300 |
| | Chi-squared | 0.540 ns | 0.149 ns | | | 1.633 ns | | | | 7.117 ns | | | | |

Note: ns = not significant. Source: Authors' calculation.

The location of Merowe Dam is unique in respect to many dams in the world; the dam is present at the heart of a desert where access to water from sources other than the Nile is very difficult. This geographical factor shows the importance of the Merowe Dam for agricultural and non-agricultural economic development through water management for irrigation or electricity generation for domestic and commercial use. The results suggest that the Merowe Dam has provided a strong and stable platform for agro-economic development in the region, in the form of electrification of irrigation systems and infrastructure for transporting goods and products to markets across Sudan and for export. These benefits include an increase in irrigated land area and adoption of modern agricultural systems with the support of the Agriculture Research Authority, which send the right message to investors to invest extensively, especially Arab investors who are the main target. Evidently, there is a sense of faith shown by many participants about the agricultural and non-agricultural economic development stimulated by the construction of the dam, especially through farming.

This research recognizes the important role played by the development of irrigation facilities in increasing productivity and income under the Merowe irrigation schemes (Tables 2 and 3). However, the facilitation of irrigation is not the only factor to have caused an increase in productivity. Hence, in the context of the results of this study, the statistical analysis highlights that increase in land size has led to significant income increase, especially for farmers. However, other factors also support the increase of income in the region, such as increase in fishing activities, market access, and emergence of new sectors, such as services.

Table 2 shows that the perception of communities on the benefits of the dam is, on balance, positive on most aspects of economic development. Chi-squared tests were conducted to examine the association between the perceived benefits across locations, which demonstrated significant differences. It illustrates significant variation ($p < 0.10$–$p < 0.01$) in perception, especially for "other purposes" and "final opinion" on the dam and its nature of contributions across three locations. This finding implies that local communities are aware of some positive economic impacts of the Merowe Dam and its supplementary projects. Their awareness appears not only confined to the visible impacts most closely related to livelihood, electricity supply, and services, but also extends to overall socioeconomic development in the region (Table 2). The performance of many large irrigation reservoirs with respect to cash generation, important crop production, and their financial returns appears to be significant, depending on availability of electricity, good infrastructure, and feasibility of irrigation schemes [16,23,31].

However, Singh [55] believed that the alleged benefits are based on the interests of the promoters of dam construction. This criticism is supported by the lack of participation and feelings about transparency shown by communities in relation to the construction of the Merowe Dam. Furthermore, lack of water for irrigation in the early stages at the relocated communities has fueled more criticisms. All the above factors, including some social conflicts, have undermined the economic benefits of the Merowe Dam. Some participants judge the Merowe Dam to be a failure and others suggest that there is no secondary benefit other than electricity generation (Table 2). Yet, this research argues that, with respect to the rural agricultural and non-agricultural economy, the Merowe Dam has clearly contributed in many areas significantly. For example, participants (6 and 17) from the local farmers' questionnaire suggested that: "The Merowe Dam helps the region economically through a stable supply of electricity and fishing opportunities for local people" and "The region has seen huge economic development across sectors, especially infrastructure, services and trade."

The construction of the Merowe Dam as a package with many supporting infrastructure projects has opened a completely new paradigm of economic development at the regional level. This idea has played a big role in driving other areas of the economy to develop (e.g., services, small manufacturing and businesses, metal workshops, construction, and tourism sectors). The following sections discuss both agricultural and non-agricultural economy through the three main areas of impact that local communities identified as the most significant for them and the region, namely electricity generation, development of modern agriculture, and other socioeconomic infrastructure development.

### 3.1. Agro-Economic Contributions

Agriculture remains as the core source of income and employment in Sudan, employing around 60% of Sudanese and making up a third of the economic sector [56,57]. The analysis of questionnaire data for this research showed that irrigation as an economic development mechanism is at the center of the development of Merowe Dam. The results indicate that there are positive perceptions about irrigation in supporting agro-economic development with no significant difference between locations regarding irrigation and agriculture. Around 60% of the participants across the locations have agreed that the development of irrigation and the agro-economy are the most positive impacts of the Merowe Dam in the region.

Bearing in mind that both Northern and River Nile States are heavily reliant on agriculture economically, such dependency on agriculture in the region added more weight to the positive perception of irrigation and increased people's expectation of the dam. Transition from traditional to modern farming might be another important factor that caused improvements in agribusiness and, to some degree, the overall Sudanese economy. The new Merowe Agriculture Development Authority has provided modern technologies for agriculture in the region to transform the old system of agriculture in the old settlements. At the new settlement, the agricultural system is completely different as farmers use modern scientific methods in all agricultural stages from preparing the land to harvesting crops. The Merowe Dam and its supportive projects helped to transform agriculture from small family-based farming to more commercial-based farming through increase of land size and introduction of new crops. However, literature also suggested some concerns about the agro-economic contributions of dams [20]. Flyvbjerg [17] and others argue that most beneficiaries of large dams are well-connected and resourced elites, who can pay for the cost of irrigation and electricity from hydropower. These assertions imply that large dams have not been beneficial to the whole society economically. However, local farmer participants #7 and #8 disagreed with this argument suggesting that: "Yes after Merowe Dam the increase in land led to increase in livestock due to availability of grazing areas and feeding grass and this led to increase in our income" (7). "Certainly there is diversity in agriculture and livestock is one area that witness increase after the dam; Northern farmers are familiar with livestock however due limitation in land there was little interest but now farmer have sufficient land, the farmer who had 5 sheep now has 50. There are around 18,632 head of lambs, 17,000 goats and 3000 head of cattle in Al-Makabrab scheme only and now northern farmers exporting to all Sudan" (8).

Table 4 presents the mean rank values of each of the indicators as ranked by the respondents on a Likert scale of 1–5 (1 = strongly disagree and 5 = strongly agree, please see Q10 of the attached questionnaire in the Supplementary S1). Table 4 shows that agro-economic indicators in the form of irrigation, types, and quality of products show some negative perceptions (overall mean rank index values ranging from 2.31–2.36, 3rd last column) as compared to other agriculture applications such as grazing areas, land size, and quality of soil (overall mean index values 2.48–2.95, 3rd last column) in the study area. This trend confirms moderate positive perceptions due to apparent increase in farm diversification including livestock.

With respect to agro-economic contributions of the Merowe Dam and its accompanying projects, indicators show significant variations across the three locations based on the Kruskal–Wallis test conducted for each indicator across locations ($p < 0.01$). Downstream communities have shown positive perception on all indicators (irrigation, types and volume of products, land size, grazing areas, quality of soil, cost of irrigation, and price of products) with some variation in the means of indicators between 3.06 and 3.63 (3rd column from the left). However, the analysis of both upstream locations shows negative perceptions toward agricultural economic contribution of Merowe Dam with some indicators scoring below 2.00 (5th and 7th columns), with the exception of land size for upstream-relocated communities (index value 2.78, 7th column top row) and quality of soil for the upstream-resident communities (index value 2.98, 5th column 2nd row). Overall mean ranking shows that land size, quality of soil, and cost of irrigation are the most influential agriculture indicators (3rd

last column, top three rows). This result is due to large strips of fertile soil formed by the annual floods, which makes the soil salt free and regenerates its fertility with regular sedimentation. In addition, generous compensation for land lost for the dam was given. Affected families were given 3 ha for each 1 ha of lost land and an additional 3 ha as a gift. However, the quality of soil at the new scheme location is lower than the quality of soil of the lost land located at the bank of the river.

**Table 4.** Ranking of farming relating economic indicators by districts.

| Sl. No. | Economic Impact of Merowe Dam Agriculture Indicators | Index Weighted by Rank of Responses | | | | | | | | Kruskal Wallis ($\chi^2$) |
|---|---|---|---|---|---|---|---|---|---|---|
| | | Downstream | | Upstream-Resident | | Upstream-Relocated | | All Region | | |
| | | Index | Rank | Index | Rank | Index | Rank | Index | Rank | |
| 1 | Land size | 3.63 | 1 | 2.45 | 2 | 2.78 | 1 | 2.95 | 1 | 39.74 *** |
| 2 | Quality of soil | 3.49 | 2 | 2.98 | 1 | 2.35 | 2 | 2.94 | 2 | 34.09 *** |
| 3 | Cost of irrigation | 3.20 | 6 | 2.28 | 3 | 1.99 | 6 | 2.49 | 3 | 46.03 *** |
| 4 | Grazing areas | 3.13 | 7 | 2.02 | 6 | 2.30 | 3 | 2.48 | 4 | 43.76 *** |
| 5 | Price of farming products | 3.06 | 8 | 2.19 | 4 | 2.12 | 4 | 2.45 | 5 | 37.64 *** |
| 6 | Volume of products | 3.21 | 5 | 2.13 | 5 | 1.99 | 5 | 2.44 | 6 | 56.49 *** |
| 7 | Irrigation | 3.27 | 3 | 1.85 | 7 | 1.97 | 7 | 2.36 | 7 | 61.53 *** |
| 8 | Types of products | 3.22 | 4 | 1.85 | 7 | 1.88 | 8 | 2.31 | 8 | 74.30 *** |

Note: *** = significant at 1% level ($p < 0.01$). Source: Authors' calculation.

The adverse effect of irrigation deficiency on upstream-relocated communities has played an important role in showing the negative impact on the agricultural economic indicators. However, respondents appreciate an increase in land size, but such an increase did not deliver positive outcomes on other agriculture related economic indicators. Insufficient irrigation caused a loss of productivity and a decrease in product quality and volume. A large majority of local farmer communities in upstream-resident location believed that the Merowe Dam had negative impacts on most of the agriculture economic indicators, with low values on all factors (mean index values 1.85–2.45, 5th column), except the quality of soil score (mean index value 2.98, 5th column, second row) which shows a positive impact. This result indicates that upstream-resident communities still suffer from relocation and loss of economic capacity. Some have also experienced neglect from the authorities due to not accepting relocation and compensation packages and, instead, relying exclusively on high-quality land of floodplain farming.

Same ranking analysis of the economic impact of Merowe Dam conducted by gender and educational level of the respondents failed to show any significant differences of ranks except in one case. Only ranking related to land-size is significantly different between gender (t = –2.677 ***). This again establishes that the differential impact of the Merowe Dam was felt differently across locations, and populations residing within the same location experienced the same effect irrespective of gender and educational level (detailed table of the analytical results not included due to limitation of space).

A Spearman rank correlation analysis of the relative ranks amongst agricultural economic indicators was performed to check whether communities' perceptions on these impacts are related and consistent. Table 5 clearly shows overwhelming significant positive and high correlation amongst all seven indicators, thereby providing confidence in the robustness and consistency of the ranking of the communities' perception from all three locations: upstream relocated, upstream resident and downstream communities. The mean values of each indicator, however, differ significantly across locations as shown by the Kruskal–Wallis test statistics presented in the last column of Table 4.

Therefore, it is important that irrigation schemes associated with mega-dams be systematic and economically driven and managed efficiently for the benefits of income and productivity increase and sustainability [23]. Examples of large irrigation dams in Brazil, India, and Turkey appear to demonstrate these benefits gained through use of technology, seed enhancement, good system of irrigation and soil preparation, storage availability for crops, and market access, which lead to a positive contribution of the dam towards economic development [23]. About 678 large Turkish dams illustrate the economic



benefits, by creating a yearly revenue worth $4 billion for the national economy [16]. The majority of farmers who participated in this study have acknowledged the benefit of using a modern and scientific agricultural system. This practice has increased productivity and income especially with an increase in land, electrification of irrigation system, introduction of new agribusiness crops, training, and support from the Agriculture Development Authority.

**Table 5.** Correlation among agricultural economic indicators.

|  | Irrigation | Types of Products | Volume of Products | Grazing Areas | Land Size | Quality of Soil | Cost of Irrigation |
|---|---|---|---|---|---|---|---|
| Irrigation | 1.0000 | | | | | | |
| Types of products | 0.6503 | 1.0000 | | | | | |
|  | 0.0000 | 0.0000 | | | | | |
| Volume of products | 0.5473 | 0.8029 | 1.0000 | | | | |
|  | 0.0000 | 0.0000 | | | | | |
| Grazing areas | 0.4947 | 0.6003 | 0.5801 | 1.0000 | | | |
|  | 0.0000 | 0.0000 | 0.0000 | | | | |
| Land size | 0.4974 | 0.5257 | 0.4848 | 0.5437 | 1.0000 | | |
|  | 0.0000 | 0.0000 | 0.0000 | 0.0000 | | | |
| Quality of soil | 0.3696 | 0.3763 | 0.3474 | 0.3591 | 0.3704 | 1.0000 | |
|  | 0.0000 | 0.0000 | 0.0000 | 0.0000 | 0.0000 | | |
| Cost of irrigation | 0.4532 | 0.5376 | 0.5099 | 0.5432 | 0.4346 | 0.366 | 1.0000 |
|  | 0.0000 | 0.0000 | 0.0000 | 0.0000 | 0.0000 | 0.0000 | 0.0000 |
| Price of farming products | 0.4514 | 0.4727 | 0.4125 | 0.4922 | 0.5512 | 0.3286 | 0.5636 |
|  | 0.0000 | 0.0000 | 0.0000 | 0.0000 | 0.0000 | 0.0000 | 0.0000 |

Note: The second row shows *p*-values ($p < 0.001$). Source: Authors' calculation.

Nonetheless, the deficiency in irrigation, which was caused by a delay in electrification of the irrigation system, to some degree, may have a negative impact on other agricultural applications, such as volume of products and prices, which led to some negative perceptions from local communities. This deficiency is a norm in Sudan's agriculture sector, suffering from low productivity, because of insufficient application of science and technology. The Aljazeera Agriculture Project clearly indicates how low productivity in recent years is one of the major contributors to why the Sudanese economy's export of farming products cannot satisfy the local needs for food nowadays. Furthermore, Varma [23,31] and Ersumer [16] believe that diversifying cash crops and the use of modern methods of agriculture have greater influence on productivity. This research recognized the important role played by an increase in irrigation area in increasing productivity and income from irrigation schemes related to the Merowe Dam.

However, irrigation area is not the only factor determining an increase in productivity. Hence, the cross-tabulation analysis presented in Table 6 highlights that income increase is significantly dependent on land size increase. The Chi-squared test result shows a significant ($p < 0.01$) positive association between land increase and income increase, implying that an increase in land size is positively associated with corresponding increase in income.

Numerous studies have supported communities' perceptions of agriculture schemes associated with dams, which have been instrumental in increasing food availability for local consumption and export, thereby leading to growth in most developing countries [16,19,23,35]. In the case of Sudan, and particularly for the Merowe area, notably from Khartoum up to the Northern States, the area has witnessed a huge development of small modern farms producing vegetables, various crops, and citrus fruits. Yet the critics of dams supporting increased food production, e.g., [7,55] suggest that, despite this inspiring economic performance, agricultural transformation has been slow, and growth is relatively weak. Singh [55] stated that crop yield is still far below potential levels, agricultural modernization is feeble and deteriorating, and the state of the agribusiness industry is still in its early stage in many developing countries.

**Table 6.** Relationship between increase in land area and income due to Merowe Dam.

|  |  | No Income Increase | Income Increased | Total |
|---|---|---|---|---|
| **Change in Land Size after Merowe Dam** | No increase in land area | 120 | 30 | 150 |
|  | Increase in land area | 39 | 111 | 150 |
| **Total** |  | 159 | 141 | 300 |
| **Chi-squared** |  | 87.796 *** |  |  |

Note: *** = significant at 1% level ($p < 0.01$). Source: Authors' calculation.

The analysis of local communities' perceptions agrees with Schultz [35], suggesting that there is a connection between increased food production and agricultural growth in supporting overall economic growth. In the case of Merowe resettlement agriculture schemes, even small differences in agricultural productivity have strong impacts on the rate and pattern of overall socioeconomic conditions in the region. Evidence of a Green Revolution in Asia and Latin America strongly suggests that agriculture can be an engine of growth in the early development process and a significant power for poverty elimination through food production and income increase [58–60]. However, this research is not able to verify if this growth has any impact on Sudan's gross domestic product (GDP) as the scope of the research focused on the regional level only and there was a as lack of information under Sudan's authoritarian regime. This research has acknowledged the impact of an authoritarian state on the research. As Merowe is a state project, the participants might have an incentive to be positive about the dam to avoid any repercussions of being negative. Despite anonymity provided by this research, as literature suggests, this is still an important factor to silence participants. Yet, to some extent, improvements in food availability and affordability support poverty reduction, which allows the socioeconomic status of Merowians to change, shifting their attention to other areas, such as education, home improvement, and even purchasing luxury goods with extra disposable income. Despite some deficiencies and challenges, the Merowe Dam has provided a strong foundation for agro-economic development in the region.

*3.2. Non-Agro-Economic Contributions*

The economic and social status of the Merowe region before the construction of the Merowe Dam depended primarily on agriculture and trade. The region has no infrastructure, which facilitates linkages between the agro-economy of rural households and other diverse areas of non-farming activities. With respect to rural economies, non-farm income enhances the value of agricultural output or income through electrification of irrigation, transportation of product to markets, and communication [61]. The above indicators have a significant impact on driving productivity, prices, and incomes, which result in an improvement in the socioeconomic status of rural communities [62]. Therefore, as this research argues, the Merowe Dam and its supplementary projects supported rural economic diversification; which virtually signifies the type of integration required between agricultural and non-agricultural economic factors at the regional and national level of Sudan. As the literature suggests, it is believed that the rural economy is not only taking advantage of non-agricultural activities, but is increasingly relying on them [63,64]. The contribution of mega-dams to other areas of economy is a controversial matter, because the focus is primarily on electricity generation and irrigation. Therefore, there is no shortage of criticism of large dams in the literature; for example, McCully [7] and Singh [55] dismiss the contributions of mega-dams to other areas of the economy and dispute alleged support for economic diversification through industrialization, creation of employment opportunities, and increased fishery from reservoirs. However, this research argues that, in respect to the rural non-agricultural economy, the Merowe Dam has clearly contributed significantly in many areas.

### 3.2.1. Electricity Generation

Sudan's economic development had been declining as a result of the loss of oil reserves and poor electricity efficiency, which suggests that the poor status of electricity supply has had a substantial impact on industrial progress [30,56]. Indeed, other factors may be responsible for the decline in Sudan's economy. However, with the introduction of the Merowe Dam, some crucial factors, such as the application of unbalanced quantity of electricity and infrastructure required to supplement other factors (e.g., agriculture in terms of crop production and agribusiness), industrial development, employment generation, communication network, and access to transportation, have been resolved in the Merowe region. A local committee member interviewee (4) elaborated on this point by noting: "There is a big shift in the region from having no electricity, no any means of communication or even televisions and see now we have communications devises and all the technology which we utilize in education at the same level as the rest of the country world. We have become more developed and willing to learn".

Electricity as a technology plays a key role in socioeconomic development, which is at the center of Merowe Dam construction. There is consensus that electricity is perceived positively, with 90% of participants across the three locations agreeing that increased electricity generation and availability has been a socioeconomic advantage. Furthermore, improvement in electricity supply ranked at the top of positive economic development impacts listed by local communities across the three locations.

The analysis of electricity produced by Merowe Dam revealed that it accounts for 1225 megawatts, thereby contributing to 50% of the total electricity produced in Sudan since 2010 according to DIU 2017 (Figures 3 and 4). Ranking of local communities' perceptions, on the Likert scale of 1–5 (1 = strongly disagree and 5 = strongly disagree), is highly positive on the availability and accessibility of electricity with an overall mean index value of 3.02 (3rd last column, top row), while the perception of the cost of obtaining electricity has a relatively moderate mean rank score of 2.38 (3rd last column, 6th row). Figure 3 further shows that domestic and agriculture share of electricity used between 2010 and 2016 is relatively higher than industrial use across Sudan including the study areas.

Certainly, the entire socioeconomic development framework of Sudan is based on the certainty and availability of electricity for agriculture, manufacturing, civil, commercial, and other purposes [39]. Figure 3 suggests that Merowe Dam development has led to growth in the use of electrical appliances domestically in Sudan, thereby leading to an increase in the domestic market. It appears that relying on agriculture as a means of economic growth and livelihood, especially after the loss of oil revenue for South Sudan, led to growth in electricity use across Sudan. Furthermore, Figure 4 shows steady contribution of the Merowe Dam to the main grid from 2010 onward. The analysis of electricity use in both Northern states shows some indication of electricity distribution by sectors.

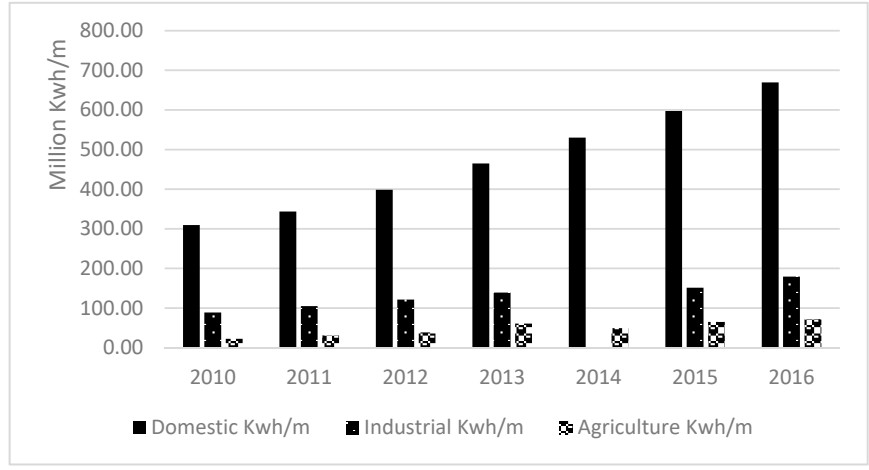

**Figure 3.** Electricity used in Sudan million KWh (2010–2016). Source: Sudanese Electricity Distribution Ltd. 2017.

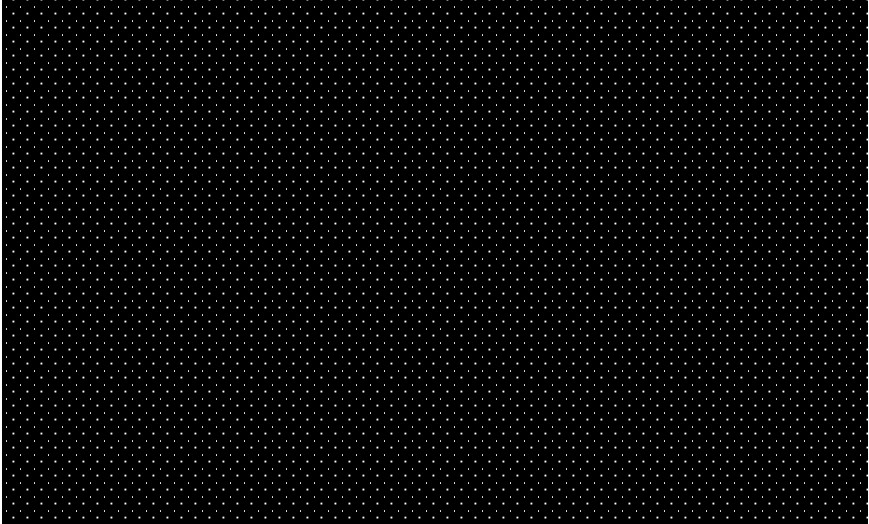

**Figure 4.** Electricity production in Sudan Merowe Dam and other sources. Source: Dam Implementation Unit (DIU) 2017.

Figures 5 and 6 show significant increase in electricity access for agriculture, industrial, and domestic uses, but there is a significant fall in electricity availability in agriculture for the years 2015 and 2016. Manufacturing has problems in the past partly because of problems of irregular supply of electricity, but since the Merowe Dam started operation in 2010, there has been improvement in industrial activities. The Merowe region has benefited slightly from multiplier influence of manufacturing growth after the Merowe Dam, especially through revival of Karimah Factory for Canning Vegetable and Fruits and establishment of Al-Nile Poultry Industry at Amri.

The Merowe Dam has significantly improved electricity supply in the region and throughout Sudan. However, the critical aspect of the Merowe Dam is the scale of such a project has caused displacement and grievances. However, significant energy needs often used to justify large-scale energy projects and the projects themselves frequently help to resolve or worsen existing social conflicts [60]. Mega-dams have been a common approach for promoting socioeconomic development, still used in many countries across Asia, Latin America, and Africa as an environmentally friendly and swift solution for energy shortages [42]. Critics in the literature suggest that capacity generation and inequitable distribution of energy generated by the Merowe Dam are overstated; many critics believe that industries and city dwellers are favored at the expense of locals [7,55]. However, perceptions of communities have challenged this view because there are significant increases in domestic use of electricity in both states in northern Sudan (Figures 5 and 6).

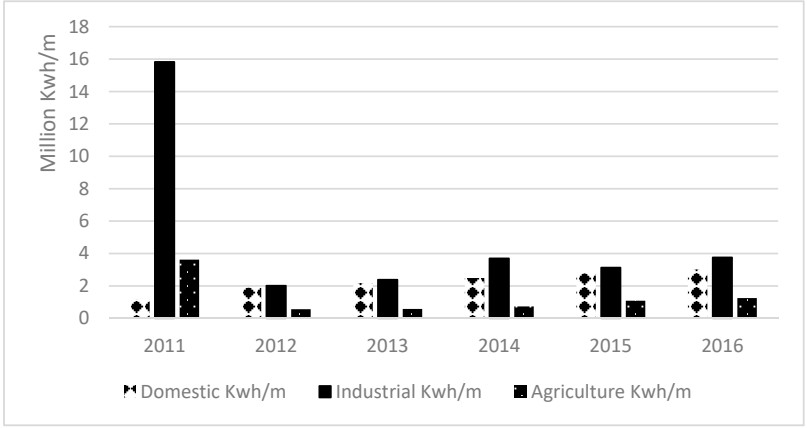

**Figure 5.** River Nile State electricity use in million KWh (2011–2016). Source: Sudanese Electricity Distribution Ltd. 2017.

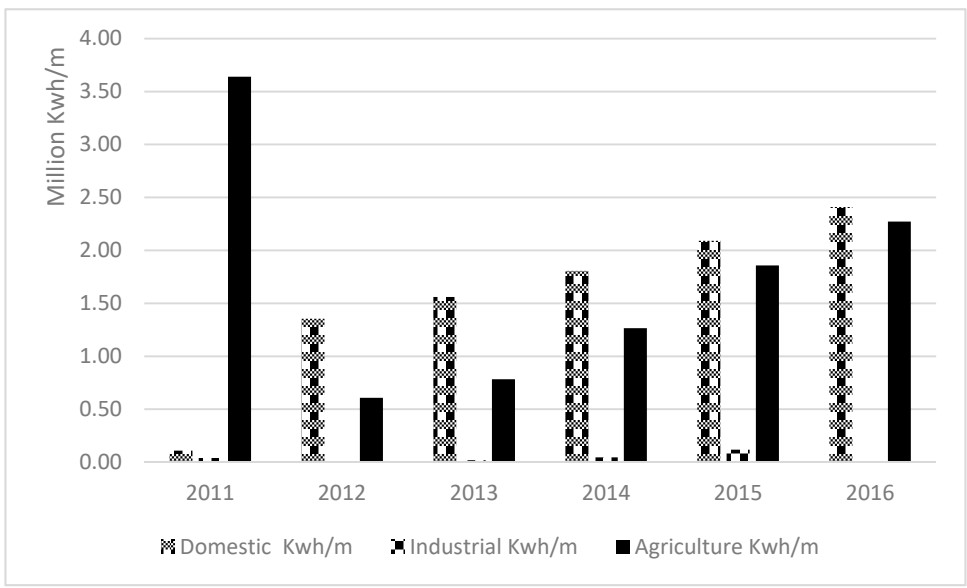

**Figure 6.** Northern State electricity use in million Kwh.Source: Sudanese Electricity Distribution Ltd. 2017.

The analyses of local participants' responses on electricity availability and the cost of access suggest different views. The results in Table 7 show significant differences across locations for both indicators, access to electricity and the cost of electricity. The Kruskal–Wallis test indicates not only significant differences across locations, but also shows that participants' positive perception on electricity availability is high but there is slight disagreement on the cost of electricity (Table 7). Viewing criticism of the Merowe Dam in relation to electricity contribution from the local participants' perspective, there is significant variation, since some participants believe that there could have been other alternatives, while others perceived that it is the only option. However, overall 54.3% believe that there is no alternative to the Merowe Dam and 45.7% believe that there are alternatives, such as renewable, nuclear, and thermal energy.

The electricity contribution of Merowe Dam drives efficiency in industries, workshops, and labs; all these sectors need electricity. Overall, there is stability in electricity supply, but there are demands to be met. However, many believe that, although the cost of production has reduced, the price of products has increased for consumers, especially after the separation of South Sudan, where the government lost oil revenue and thus relies heavily on revenue from services, especially electricity.

Certainly, the entire socioeconomic development framework of Sudan is based on the certainty and availability of energy for manufacturing, civil, commercial, and other purposes [39]. Interviewee No. 15 agreed with this notion but believed that the government of Sudan passed the cost of socioeconomic development through electricity generation to its citizens. As Sen [39] suggests, absence of access to energy in Africa results in energy poverty, as well as the fact that poor people are unable to afford access to, and use of, energy even when it is available.

This energy poverty affected Africa's socioeconomic development and citizens' quality of life. The majority of local participants acknowledge the success of Merowe Dam in improving supply and access to electricity across the region and Sudan. Other participants and interviewees expressed their appreciation of electricity that Merowe Dam produces by claiming that they have electrical appliances, such as TVs, air conditioners, fans, refrigerators, computers, and phones, and that the active time in rural Merowe region has extended beyond sunrise to sunset. A group of students from Khartoum University expressed their opinion that they have a better environment now, especially at night, because of a significant and noticeable reduction in smoke and sound that filled the atmosphere from electricity generators, which ran on fuel.

**Table 7.** Ranking non-agricultural indicators by districts.

| Sl. No. | Economic Impact of Merowe Dam Non-Agriculture Indicators | Index Weighted by Rank of Responses | | | | | | | | Kruskal–Wallis Test |
|---|---|---|---|---|---|---|---|---|---|---|
| | | Downstream | | Upstream-Resident | | Upstream-Relocated | | All Regions | | |
| | | Index | Rank | Index | Rank | Index | Rank | Index | Rank | |
| 1 | Access to electricity | 4.06 | 1 | 2.32 | 2 | 2.69 | 1 | 3.02 | 1 | 67.92 *** |
| 2 | Transportation (traveling the region) | 3.86 | 2 | 2.41 | 1 | 2.07 | 5 | 2.99 | 2 | 55.27 *** |
| 3 | Communication network | 3.74 | 3 | 2.17 | 3 | 2.63 | 2 | 2.85 | 3 | 60.62 *** |
| 4 | Infrastructure (roads, bridge, etc.) | 3.66 | 4 | 2.01 | 5 | 2.62 | 3 | 2.76 | 4 | 67.97 *** |
| 5 | Sources of income (market/jobs) | 3.30 | 5 | 1.92 | 7 | 2.07 | 7 | 2.43 | 5 | 67.76 *** |
| 6 | Cost of electricity | 2.89 | 7 | 1.99 | 6 | 2.26 | 4 | 2.38 | 6 | 33.05 *** |
| 7 | Employment generation | 2.99 | 6 | 2.01 | 4 | 2.09 | 6 | 2.36 | 7 | 38.91*** |
| 8 | Price of assets (business, land, stock) | 2.08 | 9 | 1.08 | 9 | 2.05 | 8 | 2.21 | 8 | 44.64 *** |
| 9 | Industrial development | 2.39 | 8 | 1.52 | 8 | 1.07 | 9 | 1.87 | 9 | 41.62 *** |

Note: *** = significant at 1% level ($p < 0.01$). Source: Author's calculation.

### 3.2.2. Infrastructure for Socioeconomic Development

Institutional infrastructure, in the form of education and health facilities, and the Agriculture Development Authority were built to support relocated communities to restore their lives and for locals to benefit. This development not only reduced outward migration for locals seeking education, treatment, and employment opportunities but also increased inward migration to the region. Other infrastructures, including roads, bridges, airports, and communication networks, provided huge support for goods movement, social mobilization, and trade; all these factors have changed many aspects of rural Merowians' socioeconomic conditions. Respondents identified infrastructure to be one of the means for socioeconomic development in the region.

This corresponds with the Welzel et al.'s [10] three trajectories of socioeconomic development achieved through increased productivity, market access, and increase in social mobility. These factors represent the first and second trajectories, which lead to overall improvement in living conditions, due to an increase in income subsequently leading to higher disposable income. The participants' perception on Merowe Dam and its accompanying infrastructure on the level of economic development in the study areas vary significantly as demonstrated by Kruskal–Wallis test results for each indicator across locations (Table 7). Downstream and upstream-relocated communities show consistently positive perceptions compared to upstream-resident communities with respect to all types of infrastructure and technology development. Downstream communities appear to be very positive on the impact of most of the economic indicators (mean index values 2.08–4.06, 3rd column).

The positive perception of communities was further substantiated by ending decades of isolation from the center of Sudan with the availability of roads and better transportation, electricity access, expansion of markets locally and nationally, and easier movement of goods in and out of the region. The results reveal that upstream-relocated communities experienced positive perceptions in some economic indicators, such as infrastructure, electricity, transportation, and communication scoring (mean index values 2.62–2.69, 7th column, top four rows). However, source of income, cost of electricity, and price of assets show negative perceptions (mean index values 1.07–2.07, 7th column, lower five rows). However, for upstream-resident communities, with respect to the application of infrastructure and economic indicators, perceptions are significantly negative with very low scores for all indicators (mean index values 1.08–2.32, 5th column). This situation is because upstream-resident communities have chosen not to accept government relocation and compensation packages because they preferred to stay by the reservoir in isolation from the mainstream services, and instead rely on traditional agriculture and product services away from the urban centers. Overall, the ranking of non-agricultural economic

indicators shows that after electricity access, transportation ranked at the top followed by communication network and then infrastructure, which are the main drivers of the two top indicators (3rd last column, top three rows).

The same ranking analysis of the non-agricultural economic impact of the Merowe Dam conducted by gender and educational level of the respondents failed to show any significant differences of rank except in one case. Only the ranking related to access to electricity was significantly different between gender (t = −2.587 ***). This again establishes that the differential impact of the Merowe Dam was felt differently across locations, and populations residing within the same location experienced the same effect irrespective of gender and educational level (detailed table of the analytical results not included due to limitation of space).

A Spearman rank correlation analysis of the relative ranks amongst non-agricultural economic indicators performed to check whether communities' perceptions on these impacts are related and consistent. Table 8 clearly shows high and significant positive correlation amongst all seven indicators, thereby providing confidence in the robustness and consistency of the ranking of communities' perception from all three locations.

Infrastructure has played an important role in economic development in the region. Using technology in the form of infrastructure, transportation, communication, and electricity as innovation that enhances productivity as the first trajectory of socioeconomic development based on Welzel et al.'s [10] three trajectories of socioeconomic development. Other non-farming indicators, such as source of income and price of assets, play the role of second trajectory. These feed into farming economy and enhance the overall local economy by diversifying sources of income, increasing employability, and increasing price of assets, which lead to widening of access to market and improve the socioeconomic status of people in the Merowe region [10]. Evidently, infrastructure such as roads, bridges and airports have a positive contribution economically by easing movement of goods and people across the region and to other parts of the country. This new infrastructure led to an increase in tourism to the region because many historical sites are present in Merowe region, most importantly Jabal Al-Barkal. Infrastructure plays a vital role in the formation of the International Barkal Festival, which takes place every year. Merowe Dam itself has become a tourist attraction for locals and people across Sudan. In addition, establishment of Merowe Museum, where archaeological salvages and artefacts of the Kush Kingdom are gathered and tourist attraction has increased. This economic activity provides further evidence of socioeconomic development through the tourism industry, which in turn increases income and jobs in the region.

Likewise, transportation also has a significant share of positive contribution by easing access to the region. Before Merowe Dam, it took four days to reach Merowe from Khartoum, but now with luxury high-speed buses operating in the region it takes just three to four hours, a 90% reduction in time taken to travel. Surely, communication provides the region with a vital component for business and education through use of Internet and mobile phones, whereas before the Merowe communication network, such services were limited and non-existent in some areas.

This research recognizes the influence of non-farming economic indicators on the improvement of agriculture and other areas of business in Merowe region. This evidence is clear in the form of services created and the link provided between cities through roads and bridges, in addition to the localization of health and education services. The above analysis allows this research to observe changes in socioeconomic conditions of people in the region through the Merowe Dam and its accompanying projects. These point to the positive role played by infrastructure projects improving socioeconomic status of the displaced and local people of the region. The availability of services and infrastructure that support business development and trade have led to social mobility in the region with immigration going inward into the region as compared to before the dam was constructed. "Merowe Dam brought many benefits in addition to electricity, infrastructure projects such as roads, bridges, hospitals and Merowe Technology College have provided the region with huge development opportunity" (participant 52).

**Table 8.** Correlation among non-agricultural economic indicators.

| | Industrial Development | Employment Generation | Infrastructure (Roads, Bridge, etc.) | Access to Electricity | Communication Network | Transportation (in the Region) | Sources of Income (Market/Jobs) |
|---|---|---|---|---|---|---|---|
| Industrial development | 1.0000 | | | | | | |
| Employment generation | 0.5508 | 1.0000 | | | | | |
| | 0.0000 | | | | | | |
| Infrastructure (roads, bridge, etc.) | 0.4669 | 0.6384 | 1.0000 | | | | |
| | 0.0000 | 0.0000 | | | | | |
| Access to electricity | 0.4932 | 0.5502 | 0.7374 | 1.0000 | | | |
| | 0.0000 | 0.0000 | 0.0000 | | | | |
| Communication network | 0.4888 | 0.6171 | 0.7294 | 0.7256 | 1.0000 | | |
| | 0.0000 | 0.0000 | 0.0000 | 0.0000 | | | |
| Transportation (in the region) | 0.428 | 0.495 | 0.6742 | 0.7185 | 0.6958 | 1.0000 | |
| | 0.0000 | 0.0000 | 0.0000 | 0.0000 | 0.0000 | | |
| Sources of income (market/jo) | 0.4033 | 0.5346 | 0.5695 | 0.5973 | 0.6034 | 0.5922 | 1.0000 |
| | 0.0000 | 0.0000 | 0.0000 | 0.0000 | 0.0000 | 0.0000 | |
| Cost of electricity | 0.4694 | 0.4852 | 0.4969 | 0.6467 | 0.6118 | 0.6430 | 0.5905 |
| | 0.0000 | 0.0000 | 0.0000 | 0.0000 | 0.0000 | 0.0000 | 0.0000 |

Note: The second row shows *p*-values ($p < 0.001$). Source: Authors' calculation.

The Northern region of Sudan has positively benefited economically from the Merowe Dam to some degree and its supplementary projects in the areas mentioned above and other areas, including sources of food and increase in household earnings for the overwhelming majority of local communities and farmers and non-farmers alike. This benefit has reflected on overall socioeconomic aspects of individuals and communities where this research observed a huge change in the region, although there are some social and environmental negative aspects emanating from the Merowe Dam.

## 4. Conclusions and Policy Implications

This study tackles one of the least touched issues related to the diffusion of the economic impacts of mega-dams on local communities. This study explored local communities' perception and awareness of these impacts, given that the primary goals of Merowe Dam were electricity supply and institutional infrastructure development to improve agricultural production in the region. Therefore, achievement of these goals depends on the provision of electricity for irrigation and construction of institutional infrastructure for socioeconomic improvement (i.e., roads, bridges, manufacturing, education, and hospitals facilities) in the region.

Results revealed that local communities (both farmers and non-farming residents) are fully aware of the positive impacts of Merowe Dam and its supplementary projects on the overall economy in both the agricultural and non-agricultural economy of the region. However, their awareness level remains confined within the visible impacts most closely related to their farm field and sources of livelihood and income, reflected in their ranking of various socioeconomic indicators. Both the upstream and upstream-relocated communities elicited lower mean index value of ranks for indicators, such as 'cost of irrigation', 'type and volume of products', 'land size', 'quality of soil', and 'price of products', although they have identified these as the major impact areas of the dam as well. Further analyses of relevant indicators on electricity generation and institutional infrastructure supported and validated local communities' perceptions on the positive socioeconomic impact of the Merowe Dam in the region. The approach using community perception analysis is rarely given the same priority as economic cost–benefit assessments.

However, the strength of communities' perception declines sharply on some of the agriculture related indicators, which seem to have negatively affected agro-economic development in the region. For example, deficiency in water supply for irrigation has been the main contributor to elicit lower ranking on some of the agricultural indicators. Therefore, resolving water supply issues for irrigation through effective and reliable electrification of the irrigation system at the local and regional level is needed. In addition, knowledge and the contribution of the Merowe Agriculture Development Authority on modern agricultural development seems to be limited. The local communities are not only aware of the important role of electricity and infrastructure for agricultural development in the region, but also for the entire society at large, because northern Sudan region is mainly reliant on agriculture for livelihood and income. However, results show strong emphasis on land size and relatively weaker emphasis on the quality and diversity of agricultural products.

In this paper, community perception is used to evaluate the Merowe Dam project. This approach in analyzing the dam's socioeconomic impact seems to be a useful tool in evaluating mega-dams' economic impact. This research identified that constructing the dam as a package with institutional infrastructure in addition to electricity and support for agriculture is setting a new precedent for the rural development paradigm. This can be applied in other developing countries or regions sharing similar characteristics with Merowe region. This approach has shown that a well-planned mega-dam project worked well, and benefit accrued beyond the government and elites only to reach wider communities including displaced and relocated people. However, there are some shortcomings in other areas, such as participation and consultation in decision-making regarding the dam, but these can be improved. In effect, this is a virtuous contribution to the study of dams' socioeconomic impact in the literature, especially for developing economies.

The following policy implications can be drawn from the results of this study. There is a need to rethink on the state of agriculture in the Merowe region. Investments are needed at the new settlement areas with respect to the agricultural economy, particularly by improving irrigation system through electrification, promoting crop diversity, research, development, and diffusion of modern agricultural technologies. There is also a need to strike a balance between provision of utilities and facilities, such as water supply, electricity supply, and other infrastructural facilities provided by the Merowe Dam, amongst communities in relocated, upstream, and downstream locations.

**Supplementary Materials:** The following are available online at http://www.mdpi.com/2077-0472/10/6/227/s1, Supplementary S1: Household Questionnaire Survey.

**Author Contributions:** Conceptualization, A.-N.A. and S.R.; methodology, S.R., S.E. and J.B.; formal analysis, A.-N.A.; investigation, data curation, A.-N.A.; writing—original draft preparation, A.-N.A.; writing—review and editing, S.R. and S.E. All authors have read and agreed to the published version of the manuscript.

**Funding:** This research received no external funding.

**Acknowledgments:** A special thanks goes to all the people in Sudan Merowe region and Khartoum who were actively involved throughout the research project – particularly those who were involved in the survey and interview. This work comes directly from their contributions and from their voices. To all those involved in this research from supervisors, to technical support throughout the entire process, of the study and appreciation.

**Conflicts of Interest:** The authors declare no conflict of interest.

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
