# Peer review of "Economic Contributions of Mega-Dam Infrastructure as Perceived by Local and Displaced Communities: A Case Study of Merowe Dam, Sudan"

_agriculture, doi:10.3390/agriculture10060227_

Round 1

Reviewer 1 Report

Economic Contributions of Mega-dam Infrastructure as Perceived by Local and Displaced Communities: A  Case Study of Merowe Dam, Sudan

  1. The introduction is lengthy and redundant. I would suggest having a short introduction right to the purpose of the study, a short summary of previous literature and a more detailed theoretical framework of the three paths of socioeconomic theory. In figure 1.1, the connection of the dam with the institutional infrastructures needs to be elaborated.
  2. In the method section, data was collected from seven villages. How many villages are there total? Why seven was chosen? I would like to see a copy of the questionnaire.
  3. A summary statistic of the demographics should be provided to show the characteristics of the sample and how their background affects their perceptions.
  4. A copy of the survey questionnaire can be provided in appendix.
  5. When the dam was built? This is important because the impacts may take long time to show.
  6. The article discusses the impacts of the dam and supplementary projects. However, it does not describe the supplementary projects.
  7. Line 281 “The analysis suggests….”what kind of analysis and how did you do the analysis? I expect to see more details of the analysis.
  8. Line 284 “..recently have seen some support from World Band due to their benefit to communities….” Do you mean World Bank?
  9. Table 3.1. you present the perception of people in the different locations of the dam. Do you have their demographic data? I am interested to see how the demographic factors affect their perception. Also, what do you mean here expected count? What kind of statistic method do you use? An explanation of the table is needed to help readers understand your results.
  10. Table 3.2, what these index mean and how did you get them need to be provided.
  11. Line 749 to 756, The policy implication is not based on your study results and nothing to do with the dam and related construction.
  12. How the results of this case study can be applied to other developing countries or other regions of the country?
  13. Overall, the paper should be more concise and focused and remove unnecessary details.

Author Response

Response to referee one comments

Referee 1 - The introduction is lengthy and redundant. I would suggest having a short introduction right to the purpose of the study, a short summary of previous literature and a more detailed theoretical framework of the three paths of socioeconomic theory. In figure 1.1, the connection of the dam with the institutional infrastructures needs to be elaborated.

RESPONSE: Thank you very much for your thoughtful and critical comments. We would like to confirm that we have attempted to address all of your comments to the best of our abilities. The introduction is amended and shortened substantially. It is now organised by making the purpose of the study as a starting point and made clear. The literature review shortened. The socioeconomic theory elaborated and further improvements made on the institutional infrastructure in relation to figure 1.1. Please see Lines 45 -211. The revision effectively shortened the introduction section by 80+ lines.

Referee 1 In the method section, data was collected from seven villages. How many villages are there total? Why seven was chosen? I would like to see a copy of the questionnaire.

RESPONSE: The number of villages is provided with justification of the choice of the seven villages (Please, see lines 276 -284). Furthermore, a copy of the study questionnaire survey is attached as appendix A.

Referee 1- A summary statistic of the demographics should be provided to show the characteristics of the sample and how their background affects their perceptions.

RESPONSE: A table of summary statistic by study location is provided with information on age, education, gender of respondent and agricultural activities by type activity (Please see newly added table 3.1 and lines 331-351 for explanation)

.

Referee 1- When the dam was built? This is important because the impacts may take long time to show.

RESPONSE: The starting date and completion dates provided as requested. Started in 2004 and completed in 2009. This indicates that sufficiently long time has elapsed for the dam to have clear impact (please see Line 267)

Referee 1- The article discusses the impacts of the dam and supplementary projects. However, it does not describe the supplementary projects.

RESPONSE: Brief description of the supportive projects provided (please see lines 331 and 332).

Referee 1 -Line 281 “The analysis suggests….”what kind of analysis and how did you do the analysis? I expect to see more details of the analysis.

RESPONSE: Chi-square analysis used to examine the association/difference in participants’ perceptions related to the agricultural and non-agricultural benefits brought about by Merowe Dam across three types of locations. Further elaboration made on the results of the analysis on participants’ perceptions (please see lines 356 -365)

Referee 1 - Line 284 “..recently have seen some support from World Band due to their benefit to communities….” Do you mean World Bank?

RESPONSES: Thank you. We apologise for the mistake. It is World Bank and now corrected.

Referee 1- Table 3.1. you present the perception of people in the different locations of the dam. Do you have their demographic data? I am interested to see how the demographic factors affect their perception. Also, what do you mean here expected count? What kind of statistic method do you use? An explanation of the table is needed to help readers understand your results.

RESPONSE: Based on your comment we have now prepared two new tables, Table 3.1 presenting basic demographic profile of the respondents by location (please see lines 331 – 356). Table 3.2 (previously Table 3.1) presents responses on respondent’s perception about the questions about the dam appearing on the top row of the Table. Chi-square analysis was conducted to examine association/differences in opinion on these questions by location types.

We have now removed the rows of expected counts. The expected counts actually refers to what would have been the frequencies of responses if there were NO relationship between perceptions and location types. This expected counts are produced during Chi-square analysis process as you may be aware that Chi-square statistic is defined as the Sum of (observed – expected)2/(Expected frequencies). The expected frequency of each cell in the cross-table is a product of the respective row total and column total divided by the total number of sample.  We have now explained the results more clearly and removed expected count rows to simplify the Table for the reader (please see lines 356 – 365).

We have also conducted the same analysis by Gender and Educational level (upto Secondary and above secondary) of the respondents. These results are presented in Table 3.3 (New number) and the results are discussed in the text as appropriate. We did not find any significant difference on these perceptions by gender and educational level because the experience felt within same location is same for all but vary significantly across locations (please see lines 374 – 384).

Referee 1- Table 3.2, what these index mean and how did you get them need to be provided.

RESPONSE: We have asked respondents to rank their opinions on a 5-point Likert Scale (1 = strongly disagree and 5 = strongly disagree) for each of the indicators (see Q10 – Q3 in the questionnaire). Therefore, the index is constructed by taking the average of all ranks by respondents. In theory, differences in average index values can be interpreted as valid differences in the rank of opinions across different indicators. Since, 5-point scale is used, the average index value of 2.5 means neutral response and any value above 2.5 is leaning towards ‘agree’ or ‘strongly agree’ level and vice versa. We used non-parametric Kruskal Wallis test statistic to examine significant differences in ranking of individual indicators by location types and results show significant differences throughout for all the indicators. We have provided detailed explanation of why these ranks differ by location and why some ranks were located high or low by respondents of different locations.

We would like to inform you that we have conducted same analysis by Gender and Educational level of the respondents, and again found no significant differences by gender or educational level except in one case (please see lines 553 – 559 and lines 797 - 803). 

Referee 1- Line 749 to 756, The policy implication is not based on your study results and nothing to do with the dam and related construction.

RESPONSE: The policy implications were drawn based on overall evaluation of what all those responses by the respondents from each location actually mean. We now focus only on the impact exerted in the Merowe region only and removed the part that suggests wider implication to Sudan as a whole. The clear changes brought about by provision of electricity, specific agricultural project and fishing activities and infrastructure development came about solely due to the Merowe dam. But there were criticisms on certain aspects of the dam. Therefore, the revised policy implications we have provided do flow from the results of this study and are not superficial. We have mainly emphasized improvements in agricultural modernization and the need to strike a balance in service provisions arising from the Merowe Dam to communities in relocated, upstream and downstream locations. This is because some point of contention amongst communities of different locations area about variation in the quality and effectiveness of the service provisions.

Referee 1- How the results of this case study can be applied to other developing countries or other regions of the country?

RESPONSE: The result can be applied to other countries or regions with similar characteristic to Merowe Region through development of a dam as a package of activities rather than only providing electricity and a large reservoir (Please see lines 849 – 851).

Referee 1- Overall, the paper should be more concise and focused and remove unnecessary details.

RESPONSE: We have removed a substantial part from the introduction section and also modified and/or removed unnecessary details throughout the document. The manuscript is much shorter and concise now than before.

In short, we have tried our best to accommodate all of your comments in this revised version. We strongly believe that the major revised version satisfies all of your major points and therefore suitable for publication. 

Thank you once again for your thoughtful and critical comments which have helped in substantially improving the quality of this manuscript. 

Reviewer 2 Report

Please kindly see attached the review report. 

Author Response

Referee 2- L230:  Where is Figure 1?  Figure 1.1 is the conceptual framework and Figure 2.1 is the study area but I did not see any division boundaries of the three localities, neither the upstream-relocated, downstream villages and the name of the seven villages marked in Figure 2.1.

RESPONSE: Thank you very much for your complementary comments. The mistake amended regarding figure numbers. Now there is no Figure 1. In relation to the names of the seven villages they have been added on the map at their approximate location but boundaries of location was not drawn due to lack of software that can be used to edit the map (see figure 2.1). The original map was drawn by the mapping experts of the cartographic unit of the university, which is now shut from mid-March until further notice due to CoVID-19. 

  1. I believe Table 3.1 summarized some important findings from this study. However, the authors did not do a good job in presenting their results in the table and explaining in the main context. I was very confused and don’t know how to interpret their results.
  2. How the expected count was determined/calculated? What is the ground for this expected count? Why it has to be nine for No and 91 for Yes for Electricity? Why the 9-91 ratio is the same for all three locations if you expect there would be some differences between Downstream and Upstream? Shouldn’t the differences reflect in the Expected Count and then compare with actual count from your survey respondents? Without understanding how the expected count was constructed, the comparison (L379-383) between actual and expected is less meaningful to me

RESPONSE: We apologize for the confusion created by presenting expected count. Chi-square analysis was conducted to examine association/differences in opinion on these questions by location types. We have now removed the rows of expected counts. The expected counts actually refer to what would have been the frequencies of responses if there were NO relationship between perceptions and location types. This expected counts are produced during Chi-square analysis process as you may be aware that Chi-square statistic is defined as the Sum of (observed – expected)2/(Expected frequencies). The expected frequency of each cell in the cross-table is a product of the respective row total and column total divided by the total number of sample.  We have now explained the results more clearly and removed expected count rows to simplify the Table for the reader (please see lines 356 – 365).

  1. How the perception questions summarized in Table 3.1 were framed in the survey questionnaire? Were they framed as Yes/No question for Electricity and Irrigation? And then multiple-choice questions (select only one) for Other purposes and Final opinion on Merowe Dam?

RESPONSE: These questions are not exactly framed as Yes/No questions. These are derived from the responses to a couple of open-ended questions: Q9 (what do you think are the main purposes of the Merowe dam?) and Q60 (Finally, what is your opinion on the overall building of Merowe Dam?) (Please see attached questionnaire for details of all questions). Once all different answers from these responses collated, we arrived at these classifications. Therefore, the questions are not imposed by the researcher, rather respondents were free to say what they think about the impacts of the Merowe Dam.

  1. The Chi-Square statistic was reported as a test result of what? The three distinct location groups?

RESPONSE: Yes, the Chi-square tests were conducted to see relationship between these perceptions across three locations. Based on the comments of the Referee#1 we also conducted these perception analysis by gender and educational level (upto secondary and above secondary level) of the respondent (please see new Table 3.3).

  1. The same comment applies to the expected count in Table 3.4. How is this expected count obtained and based on what? I understand a 50-50 split of 150-150 for increase/no increase in land area, that is a natural starting point to assume. But for example, why a split of 80-70 (I believe 71 is a typo) as expected count for No income increase and Income increased dichotomy groups?  Without a solid underlying theory to support the validity of expected count (i.e., nonarbitrary), the comparison of actual and expected (L477-480) is meaningless.

RESPONSE: Again we apologise for the confusion. The explanation of expected count is presented above in response to Comment 1 a. Now we have removed these expected coutns from all tables. 

  1. Please add unit to the vertical axis in Figure 3.1.

REPONSE: Added. Also, units in all other figures are added.

  1. L646-649: The authors claimed the upstream-relocated communities have experience positive perception in some economic indicators, such as infrastructure, electricity, transportation, and communication scoring (mean index values 2.80-3.00). This does not seem right to me. In Table 3.5, the index for upstream-relocated for electricity, transportation, communication and infrastructure is 2.69, 2.07, 2.63 and 2.62 respectively (see screenshot below). I felt the authors were using the All region index scoring. I will need the authors to clarify if my understanding was wrong.

RESPONSE: We apologize for the mistakes. Now, all index values are corrected and their actual location is hinted in the manuscript to correctly identify them while reading.

In short, we have tried our best to address all of your comments in this revised version. We now strongly believe that the revised version satisfies all of your major points and is now suitable for publication.

Thank you very much for your thoughtful and critical comments which have substantially improved the quality of this manuscript.

Round 2

Reviewer 2 Report

Thank you for addressing my comments. The current manuscript has been significantly improved and it is much clearer to me now. It would be ideal if the boundary of the three locations can be included in Fig. 1, but given the current circumstances, it is okay with me to forgo this request. 

I have no further questions.